# Sparsistent Model Discovery

## Abstract

Discovering the partial differential equations underlying spatio-temporal datasets from very limited and highly noisy observations is of paramount interest in many scientific fields. However, it remains an open question to know when model discovery algorithms based on sparse regression can actually recover the underlying physical processes. In this work, we show the design matrices used to infer the equations by sparse regression can violate the irrepresentability condition (IRC) of the Lasso, even when derived from analytical PDE solutions (i.e. without additional noise). Sparse regression techniques which can recover the true underlying model under violated IRC conditions are therefore required, leading to the introduction of the randomised adaptive Lasso. We show once the latter is integrated within the deep learning model discovery framework DeepMod[1], a wide variety of nonlinear and chaotic canonical PDEs can be recovered: (1) up to $\mathcal{O}(2)$ higher noise-to-sample ratios than state-of-the-art algorithms, (2) with a single set of hyperparameters, which paves the road towards truly automated model discovery.

## 1 Introduction

Mathematical models are central in modelling complex dynamical processes such as climate change, the spread of an epidemic or in designing aircrafts. To derive such models, conservation laws, physical principles and phenomenological behaviors are key. However, some systems are too complex to model with a purely bottom up approach. In such situations, and when observational data is present, automated model discovery tools are becoming increasingly more useful to derive partial differential equations (PDEs) directly from the data. For example, to model the ocean dynamics in Sanchez-Pi et al. (2020) and for embryo patterning in Maddu et al. (2020). From the mathematical point of view, model discovery of PDEs consists in finding $\mathcal{F}$ such that,

$$u_t = \mathcal{F}(1, u, u_x, u_{xx}, ...),$$

where $u_t$ is the temporal derivative of the field $u$ and $u, u_x, u_{xx}, ...$ are higher order spatial derivatives. Usually $\mathcal{F}$ is identified based on an experiment consisting of $n$ samples of the field $u$, see Brunton et al. (2016); Rudy et al. (2017); Schaeffer (2017); Raissi et al. (2017). Some recent approaches use symbolic regression to find $\mathcal{F}$, see Maslyaev et al. (2019), but so far the most popular approach to perform model discovery is by linear regression which was first introduced in Rudy et al. (2017) and consists in considering $\mathcal{F}$ as a linear combination of some candidate terms,

$$u_t = \Theta \cdot \xi,$$

where each column in $\Theta$ is a candidate term for the underlying equation, typically a combination of polynomial and spatial derivative functions (e.g. $u, u_x, uu_x$). In order to obtain a parsimonious PDE, many of the coefficients $\xi$ must be zero, motivating the use of a sparse regression method to infer the equation,

$$\hat{\xi} = \arg\min_{\xi} \|u_t - \Theta \cdot \xi\|_2^2 + \lambda \sum_i \|\xi_i\|_\rho.$$

The best subset selection is obtained for $\rho = 0$, Hastie et al. (2015). However it requires solving a nonconvex and combinatorial optimisation problem. The Lasso, Tibshirani (1996), is special in the sense that $\rho = 1$ is the smallest value of $\rho$ which leads to a convex constraint region and hence a

---

[1]Data, code and results shared on: `https://anonymous.4open.science/r/sparsistent_model_disco-56F8/`

convex optimisation problem - for which very efficient solvers exist. Furthermore, there is a large corpus of theoretical work available for the Lasso from which the model discovery community could benefit. It is anecdotaly known in the model discovery community that the Lasso does not perform very well compared to other relaxations of the original problem, see Rudy et al. (2017), Li et al. (2019), Rudy et al. (2019) and Maddu et al. (2019), however it has never been studied why. We trace back this lack of performance to the potential variable selection inconsistency of the Lasso.

Let us focus on **variable selection consistency** or **sparsistency**, a property defined as: *when the number of samples $n \to \infty$, the estimated vector $\hat{\xi}$ contains the same nonzero terms as the true vector $\xi$.* By first revisiting a geometric interpretation of the irrepresentability condition (IRC) of the Lasso in the context of model discovery, we provide some theoretical insights on Lasso's inconsistency but also on the design of the libraries generated for sparse regression based model discovery. In addition, these libraries have to be estimated using methods that introduce some deterministic noise into the sparse regression problem. We show when and why the adaptive Lasso Zou (2006) design matrices might have more chances of satisfying the IRC and introduce the randomised adaptive Lasso to perform variable selection under violated IRC. The latter is tuned by stability selection Meinshausen & Bühlmann (2010), which has been used in the past for model discovery in pure sparse regression based approaches Li et al. (2019); Maddu et al. (2019) but without any variable selection error control.

Furthermore, purely relying on sparse regression such as Rudy et al. (2017), Li et al. (2019) and Maddu et al. (2019) heavily limits to low noise and dense data sets, due to the differentiation method used to build the library (typically numerical differentiation or splines). Purely relying on deep learning, see Raissi et al. (2017), to discover $\mathcal{F}$ will result in hardly interpretable equations. In Both et al. (2021a) and Chen et al. (2020), the two problems are tackled by concurrently learning a solution of the PDE using a physics informed neural network and inferring an explicit equation by performing sparse regression on a library built by automatic differentiation. However, deep learning model discovery frameworks typically require manually tuning many hyperparameters which are sensitive to the input data: (1) the original DeepMod Both et al. (2021b) requires to tune a threshold to prune coefficients with small magnitudes, (2) PiDL Chen et al. (2020) introduces a couple of multipliers to parameter the amount of physics informed regularisation and the amount of regularisation in the sparsity estimator. Our results show how once the randomised adaptive Lasso with stability selection is integrated within the deep learning model discovery framework DeepMod, a single set of hyperparameters can be used to recover a wide variety of PDEs.

**Contributions**

- We show the design matrices used to infer the equations by sparse regression can violate the irrepresentability condition (IRC) of the Lasso, even when derived from analytical PDE solutions, i.e. without additional noise. This implies any sparse regression based model discovery framework needs to deal with highly correlated irrelevant variables with relevant ones, no matter the differentiation method used to compute the library.

- We introduce a randomised adaptive Lasso (rAdaLasso) with stability selection and error control algorithm, to recover the true underlying PDE in the presence of design matrices that are highly correlated and violate the IRC.

- By integrating rAdaLasso within the deep learning model discovery framework DeepMod, we show a wide variety of nonlinear and chaotic canonical PDEs can be recovered: (1) at higher noise-to-sample ratios than state-of-the-art algorithms, (2) with a single set of hyperparameters, paving the road towards truly automated model discovery.

## 2 THEORY

Sparse regression based model discovery sometimes fails to discover the correct underlying PDE from a data set, even when the model is present in the library and contains little noise. To illustrate this, we present a two-soliton solution[2] of the Korteweg-de-Vries (KdV) equation in figure 1(a). Considering a library $\Theta$ of $p =12$ terms, the Lasso fails to select the correct terms of the underlying

---

[2]Obtained from an analytical solution, see details in Appendix D.

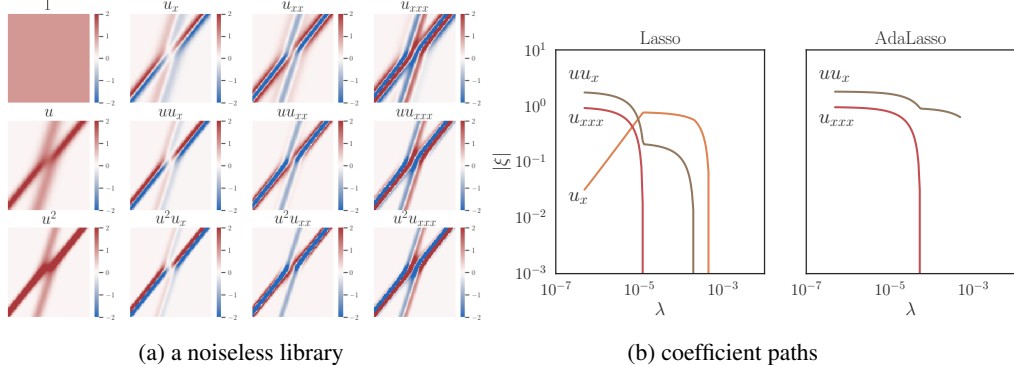

(a) a noiseless library                    (b) coefficient paths

Figure 1: *Example of PDE term selection inconsistency using the Lasso* - from a noiseless library of a two-soliton analytical solution of the Korteweg-de-Vries (KdV) equation: $u_t = -6uu_x - u_{xxx}$. In (a) the terms $u_x$ and $uu_x$ are highly correlated. In (b) no matter the regularisation, the Lasso selects the spurious term $u_x$. On the right hand side, with a proper choice of $\lambda$, the adaptive Lasso might select the true model.

PDE, see figure 1(b), even in the absence of noise. In this section, we explain why it occurs and introduce the irrepresentable condition to identify the cause.

## 2.1 ON LASSO'S INCONSISTENCY

The Lasso is known to be variable selection consistent under the *irrepresentable condition* (IRC), see[3] Hastie et al. (2015), which requires the existence of $\eta > 0$ such that,

$$\max_{j \in \mathbb{N}} ||(\Theta_T^T \Theta_T)^{-1} \Theta_T^T \Theta_{F,j}||_1 < 1 - \eta, \tag{1}$$

where $\Theta_T$ is the subset of the design matrix $\Theta$ that contains the true model and $\Theta_{F,j}$ a column of the subset of the design matrix that contains the rest of vectors. Conversely, the Lasso will not consistently select the true variables for any design matrix, even if they are present in the library.

**Geometric interpretation** it can be remarked that $(\Theta_T^T \Theta_T)^{-1} \Theta_T^T \Theta_{F,j}$ is the least squares solution of,

$$\Theta_{F,j} = \Theta_T \xi + \epsilon, \tag{2}$$

with $\epsilon$ Gaussian noise, see figure 2(a). The IRC will be violated if any of the projections of $\Theta_{F,j}$ onto the column space of $\Theta_T$ is larger than 1, meaning they lie outside the unit sphere see figure 2(b). The IRC will be trivially satisfied when $\Theta_{F,j}$ is orthogonal to the column space of $\Theta_T$, in which case $\eta = 1$: meaning irrelevant vectors are orthogonal to the relevant vectors Hastie et al. (2015). Furthermore, it becomes clear from the example on figure 2, that if the relevant vectors are orthogonal to each other then the projections of irrelevant terms onto the space of relevant vectors will be exactly the correlations and the IRC will be less likely to be violated. While it is obvious the IRC will more likely be violated in the case the data is highly correlated: correlations among irrelevant vectors will not lead to a violation of the IRC.

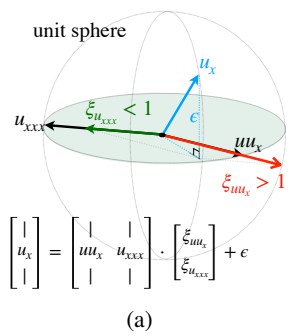

(a)

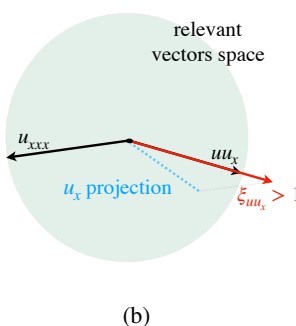

(b)

Figure 2: Example of projection that lies outside the unit sphere: the IRC is violated.

---

[3]Many formulations of the IRC can be found in literature such as the seminal work from Zhao & Yu (2006) and Meinshausen & Bühlmann (2010) - we do not pretend to be exhaustive on the matter but rather choose the one that better suits our purpose.

**A diagnostics metric** $\Delta$ based on the IRC we introduce,

$$\Delta(\Theta, T) = \max_{j \in \mathbb{N}} ||(\Theta_T^T \Theta_T)^{-1} \Theta_T^T \Theta_{F,j}||_1, \tag{3}$$

where $T$ are the vectors of the support set $S$ and $F$ on its complementary set $S_c$. In practice, the true support $S$ is unknown. Its estimation $\hat{S}$ can be obtained for example by taking for granted the result from a first variable selection. $\Delta(\Theta, \hat{T})$ can help us determining if our library $\Theta$ is sufficiently well designed: if $\Delta < 1$ we know a Lasso can distinguish $\hat{S}$ from $\hat{S}_c$, otherwise the variable selection result should be taken with caution. It is worth pointing out, that by no means if $\Delta(\Theta, \hat{T}) < 1$ we can claim $\hat{T} = T$.

## 2.2 On the adaptive Lasso's sparsistency

We have seen in the previous section that the Lasso might not be variable selection consistent when $\Delta(\Theta, T) > 1$, meaning that even if the true model is present in the library it might not be selected. Instead, we propose to use the adaptive Lasso which is known to preserve the consistency, see Zou (2006). In figure 1 we illustrate that applying it, results in the correct underlying KdV equation to be recovered. Let us give insights of why it might perform better on model discovery problems. The adaptive Lasso is a two-step estimation procedure where first an initial estimation of the coefficients is obtained to derive a weight vector $\hat{w}_i = 1/|\hat{\xi}_i|^\gamma$. We fix $\gamma = 2$ throughout this work. $\hat{\xi}_i$ is preferably obtained using a Ridge regression to handle multicollinearity. In the second step, a Lasso is applied on the weighted coefficient, $\hat{w}_i$, penalising the terms with their respective weights, i.e.,

$$\hat{\tilde{\xi}} = \arg\min_{\tilde{\xi}} \left( \frac{1}{2n} ||\partial_t u - \tilde{\Theta}\tilde{\xi}||_2^2 + \lambda \sum_{i=1}^{p} ||\tilde{\xi}_i||_1 \right), \tag{4}$$

where $\tilde{\Theta}_i = \Theta_i / \hat{w}_i$ and $\tilde{\xi}_i = \hat{w}_i \xi_i$. Let us consider the impact of the transformation on the design matrix by the adaptive weights.

**Proposition 1.** *by assuming all relevant coefficients are larger than the irrelevant ones in magnitude and $\gamma \geq 1$, then as a result of the transformation, the projection of the $j^{th}$ irrelevant vector onto the $k^{th}$ relevant vector will shrink by a factor $|\hat{\xi}_{F,j}/\hat{\xi}_{T,k}|^\gamma$ and,*

$$\Delta(\tilde{\Theta}, T) \leq \Delta(\Theta, T). \tag{5}$$

*The design matrix $\tilde{\Theta}$ will therefore have more chances to verify the IRC than $\Theta$.*

See proof on appendix A. By coming back to the example presented on figure 1, $\Delta(\Theta, T) = 1.69$ while $\Delta(\tilde{\Theta}, T) = 2e^{-14}$, which illustrates inequality 5. It is worth insisting the correlation matrices of $\Theta$ and $\tilde{\Theta}$ are identical. However, the projection of the irrelevant vector $u_x$ into the space of relevant vectors $(uu_x, u_{xx})$ becomes very small, and small enough in this case for the IRC not to be violated anymore (by replacing $\Theta$ with $\tilde{\Theta}$). This gives insights why the adaptive Lasso manages to identify the true underlying model when the Lasso would not.

## 2.3 What happens with more realistic libraries

In practice, the design matrix $\Theta$ cannot be observed, nor measured, nor derived from analytical solutions, as we did on figure 1, and has to be estimated. In classic model discovery the library $\Theta$ is typically built using splines and/or numerical differentiation. In neural network based model discovery, there will also be a non-random misfit between the output of the neural network and the data. As a result, these methods introduce non-random approximations of the higher order derivatives which can be seen as additional deterministic noise $\delta$ on top of the ground truth,

$$u_t = (\Theta + \delta) \cdot \xi \tag{6}$$

We see in practice that if the correlations introduced by $\delta$ are not too large, the true PDE can still be discovered by choosing the correct amount of regularisation for the randomised adaptive Lasso. To see the effect of the interpolation and differentiation methods, we compute $\Delta$ from 2 libraries based on a numerical solution of the chaotic Kuramoto-Sivashinsky (KS) equation (see Rudy et al. (2017))

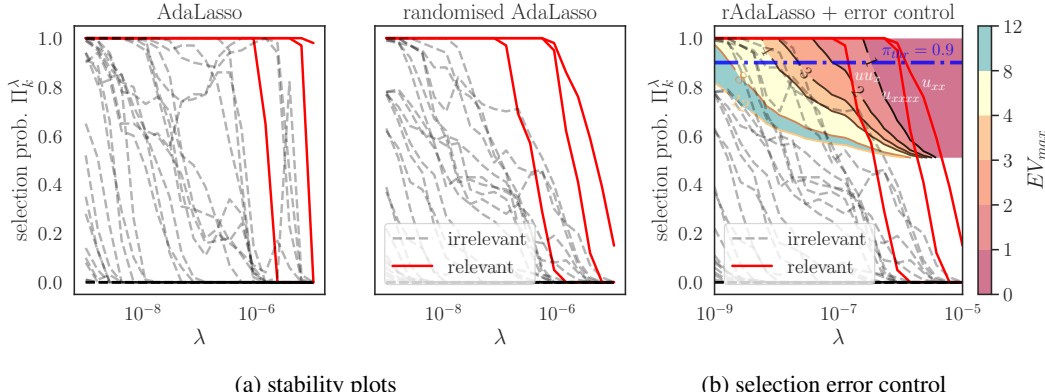

(a) stability plots          (b) selection error control

Figure 3: *Randomising the adaptive Lasso to perform model selection under violated IRC.* The library comes from Rudy et al. (2017), and was obtained by polynomial interpolation and numerical differentiation from a numerical solution of the Kuramoto-Sivashinksy equation ($u_t = -uu_x - u_{xx} - u_{xxxx}$) with additional 1% Gaussian white noise. Even $\tilde{\Theta}$ violates the IRC: $\Delta(\tilde{\Theta}, T) = 1.77$. In (a) only the randomised adaptive Lasso allows to disentangle relevant from irrelevant PDE terms. In (b), contours represent the upper bound on the selection error. The relevant PDE terms can be found by the rAdaLasso by a proper choice of the maximum number of expected false positives $EV_{max} = 2$ (at fixed minimum probability of being selected $\pi_{thr} = 0.9$).

with varying noise levels. Both libraries contain with $p = 36$ potential terms and are derived by polynomial interpolations and numerical differentiation. When no noise is added, $\Delta(\Theta_{0\%}, T) = 1.33$ and $\Delta(\tilde{\Theta}_{0\%}, T) = 7e^{-3}$ meaning the adaptive weights used by the adaptive Lasso help casting better the design matrix. However as soon as 1% noise is added, $\Delta(\Theta_{1\%}, T) = 1.38$ and $\Delta(\tilde{\Theta}_{1\%}, T) = 1.77$: the Lasso nor the adaptive Lasso would be able to select the true model. In this case, the interpolation and differentiation methods introduce enough correlations for any design matrix to violate the IRC. On figure 3, the stability plots show the adaptive Lasso would not be able to select the true model even if present in the library, no matter the amount of regularisation; this motivates the introduction of a supplementary ingredient.

## 2.4 RANDOMISED ADAPTIVE LASSO (RADALASSO)

To work under violated IRC due to correlations that are due to the underlying physical process itself and/or the interpolation and/or the differentiation method, we introduce a randomised adaptive Lasso. It is inspired by the random Lasso presented in Meinshausen & Bühlmann (2010),

$$\hat{\tilde{\xi}} = \arg\min_{\tilde{\xi}} \left( \frac{1}{2n} ||\partial_t u - \tilde{\Theta}\tilde{\xi}||_2^2 + \lambda \sum_{i=1}^{p} \frac{||\tilde{\xi}_i||_1}{W_i} \right) \qquad (7)$$

where $W_i$ is randomly selected from a beta distribution, $w \sim \beta(1, 2)$ to promote weights close to 0. Such randomisation of the regularisation is equivalent to a random rescaling of each column of the design matrix followed by an adaptive Lasso, it is therefore straightforward to solve[4]. Empirically, we can see the randomisation breaks the correlations and allows to disentangle the group of relevant from the group of irrelevant variables in a stability selection loop: see the stability plots without and with randomisation on figure 3.

**Controlling the selection error** The determination of $\lambda$ can be done using stability selection, see Meinshausen & Bühlmann (2010), which has been used in the past for model discovery Li et al. (2019); Maddu et al. (2019). However, error control has not been used and is in our opinion under-utilised: as we will see in the experimental results its hyperparameters can become data insensitive. For the sake of completeness we revisit the details of stability selection on Appendix B and recall

---

[4]It is (slightly) more computationally expensive than the adaptive Lasso with stability selection as the randomness prevents from the benefit of warm starts.

main results in the following lines. The set of variables to be selected $S_{\text{stable}}^{\Lambda^*}$ is determined using two hyperparameters: $\pi_{thr}$ is the minimum probability of being selected and an upper bound on the expected number of false positives $EV_{max}$,

$$S_{\text{stable}}^{\Lambda^*} = \left\{ k \text{ such that } \max \hat{\Pi}_k^\lambda \geq \pi_{thr} \text{ for } \lambda \in \Lambda^* \right\} \tag{8}$$

where the regularisation path is restricted by an upper bound,

$$\Lambda^* = \left\{ \lambda \in \Lambda \text{ such that, } \mathbb{E}(V) \leq \frac{q_\Lambda^2}{(2\pi_{thr} - 1)p} \leq EV_{max} \right\} \tag{9}$$

where $q_\Lambda$ is the average of selected variables. On figure 3(b), the upper bounds on the selection errors are represented using contours on top of the stability plot. The set of variables to be selected $S_{\text{stable}}^{\Lambda^*}$ is given by the terms inside a given contour line while being above the threshold $\pi_{thr}$.

To conclude this section, in the context of model discovery, sparse regression is usually performed on highly correlated data, due to the data itself and/or to the differentiation method used to estimate the library, which will tend to violate the IRC. This means that even if the true model is present in the library it might not be selected by the Lasso. As a mitigation, we introduce a randomised adaptive Lasso and show once in a stability selection loop with error control, the underlying true model can still be recovered.

## 2.5 DEEPMOD INTEGRATION

Neural network based model discovery improves the quality of the library with respect to numerical differentiation based methods, see Both et al. (2021b) . We can therefore expect the deterministic noise $\delta$ to be much smaller. To leverage such capability, we implement the randomised adaptive Lasso with stability selection and error control in the deep learning model discovery framework DeepMod[5], Both & Kusters (2020). The framework combines a function approximator of $u$, typically a deep neural network which is trained with the following loss,

$$\mathcal{L} = \underbrace{\frac{1}{n}||u - \hat{u}||_2^2}_{\mathcal{L}_{mse}} + \underbrace{\frac{1}{n}||\partial_t \hat{u} - \Theta(\hat{\xi} \cdot M)||_2^2}_{\mathcal{L}_{reg}} \tag{10}$$

The first term $\mathcal{L}_{mse}$ learns the data mapping $(x, t) \rightarrow \hat{u}$, while the second term $\mathcal{L}_{reg}$ constrains the function approximator to solutions of the partial differential equation given by $\partial_t u, \Theta$ and $(\hat{\xi} \cdot M)$. The terms to be selected in the PDEs are determined using a mask $M$ derived from the result of the randomised adaptive Lasso with stability selection and error control,

$$M_i = \begin{cases} 1 & \text{if } \tilde{\xi}_i \in S_{\text{stable}}^{\Lambda^*} \\ 0 & \text{otherwise} \end{cases} \tag{11}$$

where $i \in [1, p]$ is the index of a potential term and $S_{\text{stable}}^{\Lambda^*}$ is determined by equation (8). The coefficients $\hat{\xi}$ in front of the potential terms are computed using a Ridge regression on the masked library $(\Theta \cdot M)$. During training, if $\mathcal{L}_{mse}$ on the test set does not vary anymore or if it increases, the sparsity estimator is triggered periodically. As a result, the PDE terms are selected iteratively by the dynamic udpate of the mask $M$ during the training. In practice, this promotes the discovery of parsimonious PDEs.

## 3 EXPERIMENTS

In this section, we first show how the randomised adaptive Lasso compares with state-of-the-art sparsity estimators. Second, once within DeepMod, we compare it to the original DeepMod framework.

---

[5]The randomised adaptive Lasso promoted here, uses the Ridge and Lasso implementations from scikit-learn, Pedregosa et al. (2011). DeepMod is implemented in JAX, Bradbury et al. (2018)

Table 1: *Known challenging cases from literature.* When polynomial interpolation is used to compute higher order derivatives from noisy data, it is known that the quality of the library is going to be poor - making it challenging to discover the underlying PDE by sparse regression. For both libraries $\Delta > 1$ revealing the Lasso would not be able to recover the true support. *KS: Kuramoto-Sivashinsky.

| # | PDE | Noise | Terms | Deriv. Order | $n$ | source | $\Delta$ |
|---|---|---|---|---|---|---|---|
| 1 | KS* | $1\%$ | 36 | 5 | $250k$ | Rudy et al. (2017) | 1.38 |
| 2 | Burgers | $4\%$ | 19 | 4 | $20k$ | Maddu et al. (2019) | 1.23 |

Table 2: *Success in recovering the ground truth PDE terms for table 1 cases.* Here we reproduced the results from Rudy et al. (2017), Maddu et al. (2019) (*h* stands for heuristic) and report an additional results using the Lasso, adaptive Lasso and randomised adaptive Lasso. In case 1, PDE-FIND does find the correct terms, while it does not in case 2. In the latter, PDE-STRIDE and a randomised adaptive Lasso do, see figure 5.

| | regularisation | Case 1 | Case 2 |
|---|---|---|---|
| Lasso | $l_1$ | ✗ | ✗ |
| randomised Lasso | $l_1$ | - | ✗ |
| PDE-FIND (STRidge) | $h$ | ✓ | ✗ |
| PDE-STRIDE (IHT) | $l_0$ | - | ✓ |
| adaptive Lasso | $l_1$ | ✗ | ✗ |
| randomised adaptive Lasso | $l_1$ | ✓ | ✓ |

**Comparing with state-of-the art sparsity estimators** In order to get an idea of the performance of the randomised adaptive Lasso with stability selection and error control, we compare it to two pure sparse regression based model discovery approaches: PDE-FIND Rudy et al. (2017) and PDE-STRIDE Maddu et al. (2019). While the first is a heuristic, the latter solves a relaxation of the best subset selection ($l_0$ regularisation) using an Iterative Hard Thresholding algorithm. To make sure the comparison is fair, we compare our approach with the ones from literature using the data from the original authors of those approaches. Furthermore, we restrict ourselves to cases where the original authors have tuned their algorithms and present the cases as being hard ones, see table 1. In these cases, $\Delta(\Theta, T) > 1$, meaning they violate the IRC, see table 1. The results from the benchmark are presented in table 2. For case 1, $\Delta(\tilde{\Theta}, T) \approx 1.77$ and for case 2, $\Delta(\tilde{\Theta}, T) \approx 19$ explaining why the adaptive Lasso alone will not work in those cases. The result for case 1 is presented on figure 3. From figure 5[6], with proper tuning both the randomised adaptive Lasso as well as the Iterative Hard Thresholding (IHT) algorithm can recover the true underlying PDE of case 2. However, the computational cost of the IHT is much higher ($\times 100$) than the one of the randomised adaptive Lasso (rAdaLasso), which solves a convex optimisation problem.

**Impact of rAdaLasso in DeepMod** To quantify the impact of the proposed sparsity estimator within DeepMod we compare DeepMod with rAdaLasso and a baseline (the original DeepMod). The latter leverages a thresholded Lasso with a preset threshold of 0.1 (to cut-off small terms) and $\lambda$ found by cross validation on 5 folds. We simulate model discoveries for the Burgers, Kuramoto-Sivashinsky (KS) and two additional PDEs that introduce different nonlinearities and derivative orders: Kortweg-de-Vries (KdV), $u_t = -6uu_x - u_{xxx}$ and Newell-Whitehead (NW), $u_t = 10u_{xx} + u(1 - u^2) - 0.4$. A single set of hyperparameters is used in all cases see Appendix C. The results are reported on figure 4[7]. Our approach allows to recover all 4 PDEs without overfitting while the original DeepMod would for all, except for the KdV equation. The stability plot obtained on figure 4(b) for the KS equation can be compared to the one presented on figure 3(b): the combination

---

[6]The computational cost reported here is obtained by running the code with both the data and hyperparameters from the authors of the original work.

[7]In terms of computational cost, an epoch takes in average around $0.04$s (with $2k$ samples) on a GeForce RTX 2070 GPU from NVIDIA: discovering the KS equation takes around $90k$ epochs and around 1 hour.

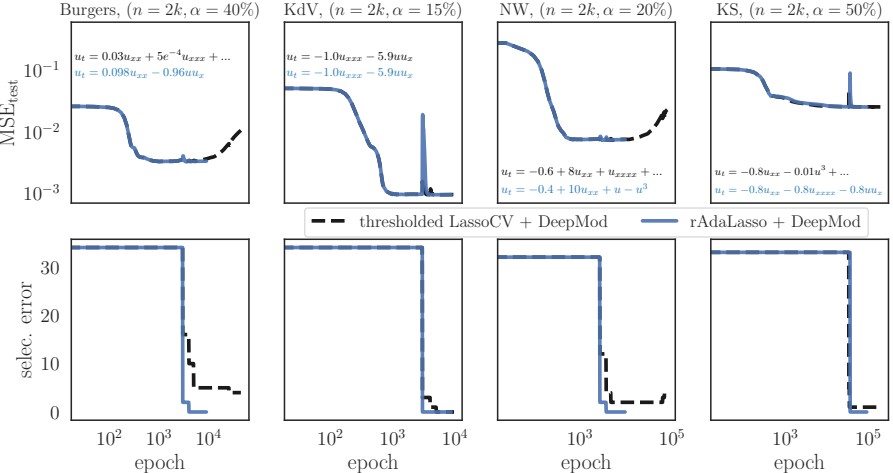

(a) MSE on test set, recovered PDEs and term selection error during learning

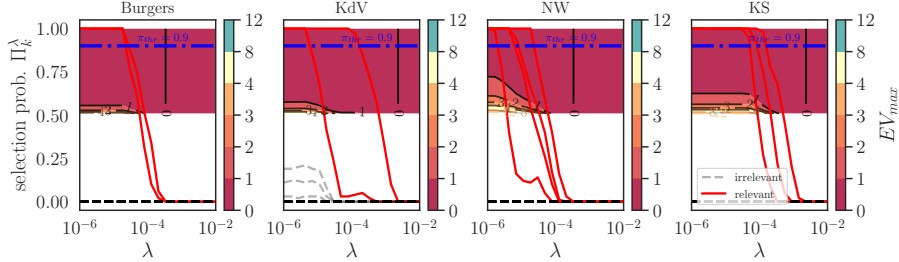

(b) rAdaLasso stability plots after DeepMod converged

Figure 4: *With and without rAdaLasso within DeepMod*. In (a), all true underlying equations are recovered when using the proposed rAdaLasso within DeepMod , from $n = 2k$ samples for varying $\alpha\%$ Gaussian white noise levels. The original DeepMod leverages a thresholded LassoCV sparsity estimator which selects spurious terms (except for the KdV example) and results in poorly generalisable PDEs (the MSE on the test set increases). In (b), the stability plots show the selected terms for all the examples have become independent (after DeepMod has converged) from the hyperparameters $\pi_{\text{thr}}$ and $EV_{max}$.

of rAdaLasso and DeepMod allow to recover the chaotic equation with greater confidence as the probability of selecting irrelevant terms is null.

**Benchmarking noise-to-sample ratios**   We compare the ratios $\Psi = \alpha/n$ where $\alpha$ is the Gaussian white noise expressed in percent of successful model discoveries of additional frameworks for the two most investigated cases in literature: Burgers and KS equations. The compared frameworks are PiDL(deep learning based thats uses the sparsity estimator of PDE-FIND, Chen et al. (2020)), S$^3$d (sparse bayesian learning, Yuan et al. (2019)), SNAPE (basis function approximations based, Bhowmick & Nagarajaiah (2021)) and R-DLGA (symbolic regression based, Xu & Zhang (2021)). The results can be found on table 3. Symbolic regression based approaches such as R-DLGA are more general than sparse regression based approaches in the sense they do need a predefined library, but do not perform as well. For the Burgers equation, the approach proposed in this work can perform the discovery at the same noise-to-sample ratio than PiDL but with a larger library ($\times 2$) and a much higher noise-to-sample ratio ($\times 100$) for the KS equation.

Some limitations of our approach are presented on Appendix F.

Table 3: *Noise-to-sample ratios (Ψ) of successful PDE discoveries from state-of-the-art frameworks.* In parenthesis, the sparsity estimator and the library size ($p$) are specified when applicable.

| framework (spar. est.) | Burgers ($p$) | KS ($p$) | source |
|---|---|---|---|
| PDE-FIND (STRidge) | $4e^{-5}$ (16) | $4e^{-6}$ (36) | Rudy et al. (2017) |
| PDE-STRIDE (IHT) | $2e^{-5}$ (19) | - | Maddu et al. (2019) |
| S$^3$d (SBL) | - | $2e^{-5}$(36) | Yuan et al. (2019) |
| PiDL (STRidge) | $2e^{-2}$ (16) | $3e^{-4}$ (36) | Chen et al. (2020) |
| SNAPE (NA) | $3e^{-3}$(NA) | - | Bhowmick & Nagarajaiah (2021) |
| R-DLGA (NA) | - | $8e^{-5}$(NA) | Xu & Zhang (2021) |
| DeepMod (rAdaLasso) | $\mathbf{2e^{-2}}$ **(36)** | $\mathbf{2.5e^{-2}}$ **(36)** | this work |

## 4 CONCLUSION

In this paper, we show that no matter the method used to compute the derivatives (numerical or automatic differentiation), the design matrices used for PDE discovery can violate the irrepresentability conditions (IRC) of the Lasso. This means irrelevant variables might be highly correlated with relevant ones. To perform model variable selection under violated IRC, we introduce a randomised adaptive Lasso (rAdaLasso). In addition, it allows to preserve a convex optimization problem and experimental results show it can select the true model in challenging cases at much lower computational cost than state-of-the-art approaches. Furthermore, once integrated in the deep learning model discovery framework DeepMod, the rAdaLasso allows to recover a wide variety of nonlinear and chaotic canonical PDEs up to $\mathcal{O}(2)$ higher noise-to-sample ratios than state-of-the-art algorithms. Finally, the hyperparameters used to perform the model discoveries are identical across experiments. These contributions pave the road towards truly automated model discovery.

Future work will focus on (1) performing discoveries by leveraging the data from several datasets using multitask learning and (2) including an evolutionary approach in the proposed framework to build the library Θ automatically.

### ACKNOWLEDGMENTS

Acknowledgements will be included in the final version.

## REPRODUCIBILITY STATEMENT

As reported in the abstract of the paper, we share the data and code on a public repository to reproduce our results. In addition,

- for figure 1, Appendix D.1 recalls the expression of the analytical solution.

- for figure 3, the repository allows to reproduce it.

- for figure 4, Appendix D.3 gives more details about the input data for DeepMod. The hyperparameters are identical across experiment and shared in Appendix C. Our repository also contains files with the requirements to reproduce the experiments using python.

The notebooks shared on the repository contain supplementary figures where one can see how well or not the modified version of DeepMod interpolates the data (especially for the chaotic Kuramoto-Sivashinsky equation).

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

## A  PROOF OF PROPOSITION 1

*Proof.* by definition for $i \in [1, p]$, $\hat{w}_i = 1/|\hat{\xi}_i|^\gamma$. We denote $\tilde{\Theta}_{T,k}$ a vector of the subset of the design matrix $\tilde{\Theta}$ that contains the relevant vectors and $\tilde{\Theta}_{F,j}$ a vector of the subset of the design matrix that contains the irrelevant vectors, then, $\tilde{\Theta}_{T,k} = \Theta_{T,k}/\hat{w}_k = |\hat{\xi}_{T,k}|^\gamma \cdot \Theta_{T,k}$ and $\tilde{\Theta}_{F,j} = \Theta_{F,j}/\hat{w}_j = |\hat{\xi}_{F,j}|^\gamma \cdot \Theta_{F,j}$ where $k$ is some column index of $\tilde{\Theta}_T$ and $j$ of $\tilde{\Theta}_F$. The projection of an irrelevant vector onto the space of relevant vectors in the least-squares sense $\Theta_{F,j} = \Theta_T \xi + \epsilon$ can be decomposed into the space of relevant vectors, $\Theta_{F,j} = \sum_k \xi_{F,j,k} \Theta_{T,k} + \epsilon$, where $\xi_{F,j,k}$ are the projections of a given irrelevant vector onto the relevant space. Using such decomposition the irrepresentability condition becomes, $\max_j \sum_k |\xi_{F,j,k}| < 1 - \eta$. By applying the transformation due to the adaptive weights we get, $\tilde{\Theta}_{F,j} = |\hat{\xi}_{F,j}|^\gamma \sum_k \xi_{F,j,k} \cdot \frac{1}{|\hat{\xi}_{T,k}|^\gamma} \cdot \tilde{\Theta}_{T,k} + \epsilon \cdot |\hat{\xi}_{F,j}|^\gamma$, which can be reduced to $\tilde{\Theta}_{F,j} = \sum_k \left|\frac{\hat{\xi}_{F,j}}{\hat{\xi}_{T,k}}\right|^\gamma \xi_{F,j,k} \cdot \tilde{\Theta}_{T,k} + \epsilon \cdot |\hat{\xi}_{F,j}|^\gamma$. Now if we assume that since $\hat{\xi}_{T,k}$ is a relevant term and $\hat{\xi}_{F,j}$ is an irrelevant one: $|\hat{\xi}_{T,k}| > |\hat{\xi}_{F,j}|$. If in addition, $\gamma \geq 1$, then, $0 \leq |\hat{\xi}_{F,j}/\hat{\xi}_{T,k}|^\gamma < 1$, which leads to,

$$\sum_k \left|\frac{\hat{\xi}_{F,j}}{\hat{\xi}_{T,k}}\right|^\gamma |\xi_{F,j,k}| \leq \sum_k |\xi_{F,j,k}| \tag{12}$$

which is equivalent to the inequality 5. □

## B  STABILITY SELECTION WITH ERROR CONTROL

With stability selection, variables are chosen according to their probabilities of being selected with a warranty on the selection error, Meinshausen & Bühlmann (2010). The first step consists in finding the probability of a variable $k$ of being selected under a data perturbation: let $I_b$ be one of $B$ random

sub-samples of half the size of the training data drawn without replacement. For a given $\lambda$ an estimation of the probability of $k$ being selected is given by,

$$\hat{\Pi}_k^\lambda = \frac{1}{B} \sum_{b=1}^{B} \begin{cases} 1 & \text{if } f_k^\lambda(I_b) > 0 \\ 0 & \text{otherwise} \end{cases} \tag{13}$$

where $f_k^\lambda(I_b) = ||\hat{\xi}_k^\lambda(I_b)||_1$ for the randomised adaptive Lasso. Second, by computing the probabilities of being selected over a given range of $\lambda$'s the stability paths can be obtained, see figure 3. That range is denoted $\Lambda = [\epsilon\lambda_{max}; \lambda_{max}]$, where $\epsilon$ is the path length and $\lambda_{max}$ is the regularisation parameter where all coefficients $\hat{\xi}$ are null. In Meinshausen & Bühlmann (2010) derive an upper bound on the expected number of false positives $\mathbb{E}(V)$, that can help determining a smaller $\Lambda$ region where a control on the selection error can be warrantied. By fixing this bound to $EV_{max}$, the regularisation region becomes, see figure 3,

$$\Lambda^* = \left\{ \lambda \in \Lambda \text{ such that, } \mathbb{E}(V) \leq \frac{q_\Lambda^2}{(2\pi_{thr} - 1)p} \leq EV_{max} \right\} \tag{14}$$

where $\pi_{thr}$ is the minimum probability threshold to be selected and $q_\Lambda$ is the average of selected variables. We propose here to approximate $q_\Lambda$ by $\hat{q}_\Lambda = \frac{\sum_b |S_b|}{B} = \sum_k \hat{\Pi}_k^\lambda$. Finally, the set of stable variables with an upper bound on the expected number of false positives is,

$$S_{\text{stable}}^{\Lambda^*} = \left\{ k \text{ such that } \max \hat{\Pi}_k^\lambda \geq \pi_{thr} \text{ for } \lambda \in \Lambda^* \right\} \tag{15}$$

## C   A SINGLE SET OF HYPERPARAMETERS

**Stability selection**   expected number of false positives upper bound $EV_{max} = 3$, number of re-samples $B = 40$ and the minimum probability to be selected $\pi_{thr} = 0.9$.

**Library**   consists of polynomials and partial derivatives up to the fifth order leading to a library size of $p = 36$ potential terms: $\{1, u_x, u_{xx}, ..., u_{xxxxx}, u, uu_x, ..., uu_{xxxxx}, ..., u^5, u^5 u_x, ..., u^5 u_{xxxxx}\}$.

**Neural network architecture & optimiser**   NNs are 4 layers deep with 65 neurons per layer and sinus activation functions with a specific initialisation strategy, see Sitzmann et al. (2020). The NNs are trained by an Adam optimiser with a learning rate of $5 \cdot 10^{-5}$ and $\beta = (0.99, 0.99)$.

**Randomness**   Seeds are identical across datasets: (1) to initialise the NNs and (2) generate the noise vectors.

## D   ABOUT THE DATA

This appendix provides data source details to reproduce the examples of this paper.

### D.1   ANALYTICAL LIBRARY OF THE KDV EQUATION

The **Kortweg-de-Vries (KdV)** $u_t = -6uu_x - u_{xxx}$ PDE analytical solution for 2 travelling solitons is,

$$u(x,t) = 2(c_1 - c_2) \cdot \frac{c_1 \cosh(\sqrt{c_2}\xi_2/2)^2 + c_2 \sinh(\sqrt{c_1}\xi_1/2)^2}{\left( (\sqrt{c_1} - \sqrt{c_2}) \cosh[(\sqrt{c_1}\xi_1 + \sqrt{c_2}\xi_2)/2] + (\sqrt{c_1} + \sqrt{c_2}) \cosh[(\sqrt{c_1}\xi_1 - \sqrt{c_2}\xi_2)/2] \right)^2}$$

where $c_1 > c_2 > 0$, $\xi_1 = x - c_1 t$ and $\xi_2 = x - c_2 t$. 40 points equally distributed such that $x \in [-5, 12]$, 50 points equally distributed such that $t \in [-1, 2]$ and $c_1 = 5, c_2 = 2$. By automatic differentiation we obtain the 12 terms library: $\{1, u_x, u_{xx}, u_{xxx}, u, uu_x, uu_{xx}, uu_{xxx}, u^2, u^2 u_x, u^2 u_{xx}, u^2 u_{xxx}\}$.

## D.2 Libraries from splines/numerical differentiation

**Burgers,** $u_t = \nu u_{xx} - u u_x$, shared on the github repository mentionned in Maddu et al. (2019). The solution here is very similar to the one obtained using the analytical expression below using Dirac delta initial conditions.

**Kuramoto-Sivashinky (KS),** $u_t = -u u_x - u_{xx} - u_{xxxx}$, shared on the github repository mentionned in Rudy et al. (2017).

## D.3 Input data for deep learning experiments

We generate numerical solutions from several equations, on top of which we add $\alpha$ Gaussian white noise,

$$u_{\text{noisy}} = u + \alpha \cdot \sigma(u) \cdot Z \tag{16}$$

where $Z \sim N(0,1)$. The following PDEs are considered:

**Burgers,** initial condition: Dirac delta, analytical solution,

$$u(x,t) = \sqrt{\frac{\nu}{\pi t}} \cdot \frac{(e^{\frac{A}{2\nu}} - 1)e^{\frac{-x^2}{4\nu t}}}{1 + \frac{1}{2}(e^{\frac{A}{2\nu}} - 1)\text{erfc}(\frac{x}{\sqrt{4\nu t}})}$$

where $A$ is a constant and $\nu$ is the viscosity, $\nu = 0.1$, $A = 1$ and 40 points equally distributed such that $x \in [-2,3]$, 50 points equally distributed such that $t \in [0.5,5]$.

**Kortweg-de-Vries (KdV),** see subsection D.1.

**Newell-Whitehead (NW),** $u_t = 10u_{xx} + u(1 - u^2) - 0.4$, numerical solution using a finite differences solver and the following initial condition:

$$u(x,0) = \sum_{i=1}^{3} \alpha_i \sin(\beta_i \pi x)$$

where $\alpha_i$ and $\beta_i$ are constants. 40 points equally distributed such that $x \in [0,39]$, 50 points equally distributed such that $t \in [0,1.96]$ and $\alpha_1 = 0.2, \alpha_2 = 0.8, \alpha_3 = 0.4, \beta_1 = 12, \beta_2 = 5, \beta_3 = 10$.

**Kuramoto-Sivashinky (KS),** see subsection D.2. 2000 samples are randomly drawn from a subset of the dataset, details can be found on our github repository, see note 1.

## E Additional Results

**Stability plots for case 2 comparison** In this case the performance of PDE-STRIDE and rAdaLasso are compared on figure 5.

**DeepMod interpolations for the experiments** see figure 6.

## F Limitations of the approach

**Incomplete library** An obvious limitation of the approach is that the library of potential PDE terms must contain the true underlying PDE terms. Typically this can be diagnosed as we report (in TensorboardX) the mean square error on a test set which allows to verify if the discovered PDE generalizes well or not.

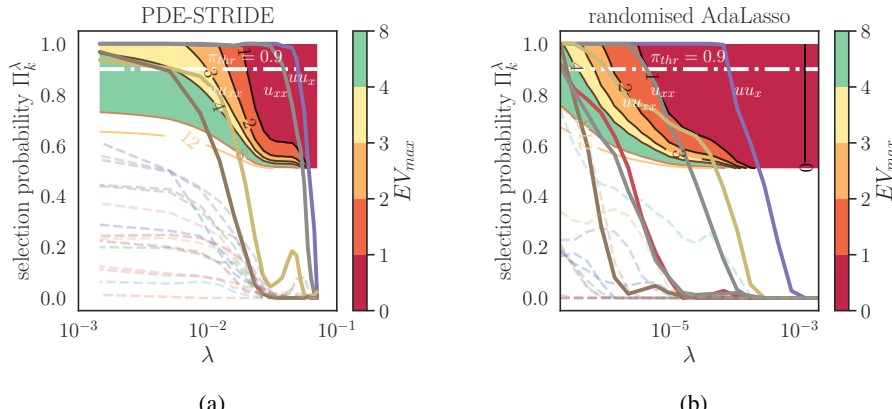

(a)             (b)

Figure 5: *Comparing PDE-STRIDE and the randomised adaptive Lasso selection performance on a challenging case*: recovering the Burgers' equation from a library built using polynomial interpolation from a dataset with $4\%$ noise Maddu et al. (2019). In (a), PDE-STRIDE solves a relaxation of the best subset selection ($l_0$ regularisation) using an Iterative Hard Thresholding algorithm. In (b), the stability plot for the randomised adaptive Lasso. The true underlying PDE can be recovered by both methods by a proper tuning of the error selection: $EV_{max} = 2$. However, the computational cost to run PDE-STRIDE is a couple orders of magnitude higher ($\approx 122s$) compared to the one of for the randomised adaptive Lasso ($\approx 1.30s$).

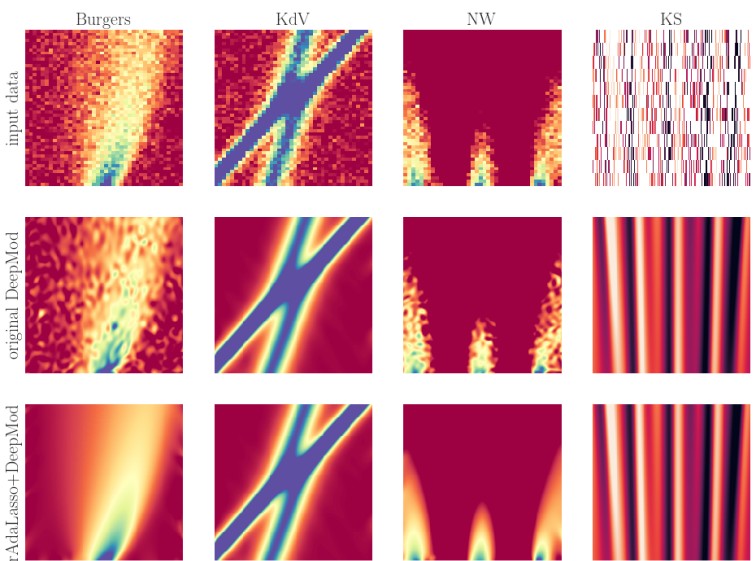

Figure 6: *DeepMod interpolations for the experiments described in the main text.*

**Non-unique solutions**    If the discovery problem has non unique solutions, our approach will propose one that might not be the one we were looking for. So far, we have devised two cases in which the solutions are not unique and we present them in the next paragraphs. It is known that the solution of a single soliton from the KdV equation is also a solution of the simpler travelling wave equation: $u_t = -cu_x$, see Rudy et al. (2017). So trying to discover the underlying equation from an analytical of KdV with a single soliton like,

$$u(x,t) = \frac{c}{2\cosh(\sqrt{c}\frac{x-ct}{2})} \tag{17}$$

where $c > 0$, will result in the discovery of $u_t = -cu_x$. We obtained a similar result with our approach while trying to find the underlying equation from data generated by a solution of Fishers equation ($u_t = u_{xx} + u(1 - u)$) from Ma & Fuchssteiner (1996),

$$u(x,t) = \left(1 + (\sqrt{2} - 1)e^{\frac{-\sigma(x+2\lambda t)}{2}}\right)^{-2} \tag{18}$$

where $\sigma = \lambda - \sqrt{\lambda^2 - 1}$ and $\lambda = \frac{5}{2\sqrt{6}}$. For both analytical solutions $x$ and $t$ play a symmetric role explaining why $u_t = k \cdot u_x$, where $k$ is some constant.

**Coefficient bias in chaotic systems**    for the successful discoveries of non chaotic PDEs, the coefficient mean errors are typically below ($< 2\%$). For the chaotic Kuramoto-Sivashinsky PDE, we show that with $50\%$ noise and a fraction of the data we can recover the terms of the PDE with half the mean error on the coefficients magnitudes with respect to PDE-FIND. However, chaotic systems are sensitive to their initial conditions and the coefficients errors ($20\%$) of the discovered PDE are very large with respect to the ground truth. As a result, the discovered PDE cannot be used for predictions but can be used to identify the underlying physical processes.

