# OpenReview forum: "Sparsistent Model Discovery"
_ICLR.cc/2022/Conference — ICLR 2022 Submitted_

### Official Review · Reviewer_1UJP · 2021-11-01

**Correctness:** 4
**Technical Novelty And Significance:** 2
**Empirical Novelty And Significance:** 3
**Recommendation:** 3
**Confidence:** 3

**Main Review:**

Post-response update. The author response clarified my technical comments. My opinion has not changed: the paper still has insufficient empirical exposition and experiments.

-----------

The paper is well written, and presents both the problem and its solution relatively clearly. The IRC problem is well presented and motivated, but I would have liked to see more exposition on the context. How does IRC connect with collinearity (or other types of identifiability issues) and identifiability in general? Giving more context would clarify the scope of the paper.

There is considerable literature on SINDy style methods, and the field is already mature and well-developed. The SINDy papers usually have extensive experimental sections: the original SINDy paper did 6 systems with and without noise, and later papers have expanded on this.

Since the new lasso method should be universally applicable to any SINDy type method, one then naturally expects to see experiments on applying the new radalasso to most systems already considered in the literature, and showing improvement. One would expect to see ablation studies comparing the performance as a function of data size, noise levels or system complexity/stifness to explain in which situations or regimes the old/new lasso method fails/succeeds.

Unfortunately, the paper does not consider ODEs at all; limits the experiments to few PDEs; and doesn’t do any ablation studies. This limits the impact of this paper significantly. (If one only considers PDEs, the title is then overly broad as well!). The DeepMod experiment is interesting, but requires explaining the resulting difference by eg. varying the noise or data size. The results are also a bit ambiguous, it seems that selection error is similar for the two methods, yet the identified systems seem quite different. The experiment of table 3 shows good results, especially with KS.

The two or four systems considered are not sufficient to show that the new method consistently outperforms SOTA.

The paper needs to consider more previous systems, and show that (i) it performs effortlessly in “easy” systems, (ii) it performs well for many “difficult” systems, and (iii) it works for both ODEs and PDEs and within neural methods, (iv) how it performs on various levels of noise and data size. Also studying how the lasso behaves under varying system "complexities" would be useful exposition to the problem.

Technical comments
* Fig 1: This is excellent figure, and shows the adalasso performs well. However, the introduction stated the goal of the paper to show that lasso fails even if one takes n -> infinity. This figure should then also present the lasso/adalasso behavior under different data counts: does the problem persist also in high-data regimes?
* Sec 2.2. claims that ridge regression “handles” multicollinearity. It is unclear what multicollinearity means in this context, what it means to “handle” something, and why ridge handles it. Ridge regression finds dense solutions, and I fail to see how this handles collinearity. Please clarify.
* Eq 4 is vague. We optimise \tilde\xi, yet \tilde\xi is defined to be w\xi. So then one needs to either optimise w or \xi instead (or both). I assume that one ultimately is optimising \xi, since w’s are found as preprocessing. Please clarify
* Can you clarify if the adaptive lasso is off-the-shelf, or a some new variant of it.
* Prop 1: What does “relevant coefficient” mean? Does this refer to \xi's or something else?
* Sec 2.3. claims the design matrix has to be estimated. I’m confused: isn’t the point of SINDy type models that one enumerates “all" combinations of the covariates and derivatives, and the \xi then selects from them. What does the “estimation” here refer to? What’s the role of the splines here? Is this referring to non-stationary bases?
* Eq 6: I don’t understand why one corrupts the underlying terms, instead of corrupting the coefficients \xi. Generally this is confusingly written, and its unclear what the neural networks are doing here. They were not discussed previously, and NNs are not present in eq 4. Please this section, and show how the neural network plays into equation 4.
* Fig4: what is the x-axis of top row? What do the equations in top row mean? What is k? What is select. error (please define).

**Summary Of The Paper:**

The paper discusses the unidentifiability of lasso-based differential equation discovery, and proposes to use a more stable lasso variant (randomized adaptive lasso), which can more accurately identify PDE coefficients. The method is well-motivated and adresses a major bottleneck in differential discovery. The method is correctly presented, and the initial results are excellent.

**Summary Of The Review:**

In summary the method is a great contribution to PDE discovery methods, but requires a much more comprehensive experimental section to convincingly demonstrate its benefits under different settings and demonstrate insight into its behavior. Currently the paper feels premature for publication, but no doubt will become an excellent paper in future.

---

> ### Author Response · Authors · 2021-11-14
> **point by point**
>
> First of all we would like to thank the reviewer for her/his very insightful comments that will help without no doubt to improve our work.
>
> 1. Fig 1: the problem does persist with more data, but we agree with the reviewer it is a great idea to report it somehow.
> 2. Sec 2.2. about multicollinearity in this context. This is imprecise from our side, we meant collinearity - is that clearer? We are not sure we understood your concern, could you reformulate it?
>
> 3. Eq 4 is vague. We realize our notation is misleading. We optimise \tilde\xi and then we can obtain \xi using the weights and \tilde\xi.
>
> 4. the adaptive Lasso is off the shelf, see: Hui Zou. The adaptive lasso and its oracle properties. Journal of the American statistical association, 101(476):1418–1429, 2006. (cited in the paper)
>
> 5. Prop 1: A relevant term is a \xi that belongs to the true support (ground truth).
>
> 6. Sec 2.3. To be able to concatenate all potential terms in the desgin matrix (library) one needs to compute these terms first. If we want to include derivatives as potential terms selected by the \xi's then these derivatives need to be computed somehow. One typically has access to the field u but not u_t nor u_x, etc. This is why we say no matter the derivation technique used, the library has to be estimated. SINDy or PDE-FIND typically leverage numerical differentiation (splines, polynomials, finite differences, etc.) while DeepMod leverages neural networks and automatic differentiation. Does this help?
>
> 7. Fig4: what is the x-axis of top row? The same as the bottom row, both plots are synchronized.
> What do the equations in top row mean? We realize this plot is not clear and needs improvement: those equations are the ones recovered either by DeepMod with and witout rAdaLasso.
>
> Thank you for the encouragement !

---

### Official Review · Reviewer_dRBM · 2021-11-02

**Correctness:** 3
**Technical Novelty And Significance:** 2
**Empirical Novelty And Significance:** 3
**Recommendation:** 5
**Confidence:** 3

**Main Review:**

Strengths:

1) A drawback of sparse regression based equation inference is exposed: the dictionaries can violate the irrepresentability condition (IRC) of Lasso, even when derived from analytical PDE solutions.
2) To alleviate the drawback, a randomised adaptive Lasso (rAdaLasso) is proposed and intergated within the DeepMod framework with promising empirical performance.

Weakness:

1) (Technical novelty). The technical novelty of the proposed rAdaLasso seems incremental to random Lasso. It is suggested to tell the readers what is really new in the proposal.
2) (Theoretical depth). The proposed rAdaLasso is shown to perform better when IRC is violated. However, the theoretical analysis of the underlying machenism is insufficient. It is possible to theoretically characterize how better the proposed rAdaLasso can perform?

**Summary Of The Paper:**

This manuscript studies the variable selection consistency in model discovery of PDEs. A randomised adaptive Lasso (rAdaLasso) with stability selection and error control algorithm is proposed to recover the true underlying PDE in the presence of design matrices
that are highly correlated and violate the IRC. Better model recovery performance is obtained by integrating rAdaLasso within the deep learning model discovery framework DeepMod.

**Summary Of The Review:**

The problem of variable seclection consistency of model discovery in PDEs is studied by first pointing out that sparse regression may fail with violated IRC and then proposing a randomised adaptive Lasso to tackle it. To me, this work is interesting. However, the proposed rAdaLasso seems not quite novel, and its relevant theroetical analysis seems a bit weak.

---

> ### Author Response · Authors · 2021-11-14
> **point by point**
>
> First of all we would like to thank the reviewer for her/his very insightful comments that will help without no doubt to improve our work.
> 1. We agree with the reviewer we need a better explanation of why it seems to work better than the random Lasso.
> 2. There seems to be some agreement with the previous reviewer about the theoretical analysis in our work. We thank the reviewer for this comment and will work in this direction in the future before re-submitting.

---

### Official Review · Reviewer_kL72 · 2021-11-03

**Correctness:** 2
**Technical Novelty And Significance:** 2
**Empirical Novelty And Significance:** 3
**Recommendation:** 5
**Confidence:** 3

**Main Review:**

The paper is generally well written and the problem of variable selection when modelling dynamical systems is well motivated.
I have following reservations about this:
- Diagnostic tool developed in the paper needs $\Theta_T$. It seems to me that It does not have much application beyond the PDEs where we know true variables.
- In proposition 1, it's not clear why the assumption that relevant coefficients are larger than irrelevant coefficients. Since this assumption is the basis of the proof, it's important to clarify when this assumption will be met.
- The onus on the choice of randomisation used in rAdaLasso ($\beta (1, 2)$.) is not very clear to me. How doe the performance change if I change this distribution and how does that affect the performance of the rAdaLasso?
- In Figure 4, authors claim MSE on test in has poor generalisation. I don't understand why the blue curve (corresponding to the rAdaLasso) stops after certain number of epochs in 3 examples. If we only compare the performance of rAdaLasso when they are trained for same number of epochs, their performance is comparable.
- Finally empirical evaluation in a more realistic example where one doesn't know the true support of the design matrix. I understand that computing the selection error on such examples is difficult, one can still look at the MSE loss for comparison.



**Summary Of The Paper:**

This paper deals with the problem of model discovery based on sparse regression. Authors propose a new randomised adaptive lasso for that selects the terms to be used in the PDEs selection that is used in conjunction with Deep learning framework. Authors then experimentally verify on 4 PDEs that their method perform better for model selection and at least as well when looking at MSE loss.

For a detailed summary, authors first verify the violation of IRC which is defined as in eq (1) for Lasso on an analytic PDE (Korteweg-de-Vries (KdV) equation). As their first contribution, authors develop a diagnostic metric that indicates the violation of Lasso when one know the true support of the design matrix.
In proposition 1, authors show that under certain assumptions on weights, the adaptive Lasso, where one reweighs the design matrix has better diagnostic score than Lasso.
Since the design matrix is typically approximated, if we incur a deterministic error term, authors empirically show that adaptive Lasso is also susceptible to the violation of the IRC. In lieu of this experiment, authors propose randomised adaptive Lasso, where each column of the design matrix of adaptive Lasso is randomly scaled with $\beta(1,2)$. Author integrate it in deep learning model discovery framework and empirically verify the performance of of rAdaLasso for 1) MSE loss and 2) model selection error.

**Summary Of The Review:**

Authors proposed a heuristic algorithm, randomised adaptive Lasso, that is evaluated empirically on 4 analytic PDEs. In those specific cases authors show success of their algorithm in model selection problem. Having said that I still recommend to reject this submission because I think the empirical evaluation is limited to PDEs where we know the True coefficients and generalisation on more difficult problems in unknown. And finally, the limited theoretical justification behind the work has some strong assumptions which are not sufficiently discussed in the paper.

---

> ### Author Response · Authors · 2021-11-14
> **point by point**
>
> First of all we would like to thank the reviewer for her/his very insightful comments that will help without no doubt to improve our work.
>
> 1. Indeed, the diagnostic tool requires knowing the true support. It is useful to characterize how hard a problem is for a model discovery machine. It is up to our knowledge computed for the first time on model discovery data and could help setting up benchmark datasets for the model discovery community on hard cases.
>
> 2. Assumption in proposition 1. The main role of proposition 1 is to give some intuition of when the adaptive Lasso will work better than the Lasso. This being said, our assumption seems reasonable when the relevant coefficients of the underlying PDE are large, meaning the physical processes happen at similar scales. We will look further into this.
>
> 3. Choice of randomisation used in rAdaLasso. We agree this is poorly motivated and should be improvedn, we will work on this.
>
> 4. All curves stop when DeepMods convergence criterion is met. The convergence criterion is that the coefficients of the discovered PDEs do not vary up to a certain precision for some fixed amount of epochs.
> We agree that the performance in terms of MSE seems to be comparable for the same number of epochs. This could really happen if the problem at hand does not have unique solutions: the solution of a given PDE can be the solution of many different PDEs. In the presented cases, a zoom on the MSE will show it is lower when the true PDE is found. We agree this is not clear enough and needs to be addressed.
>
> 5. Indeed, it would be great to have some experimental data to work on, we are working on it.

---

### Official Review · Reviewer_pXwU · 2021-11-06

**Correctness:** 2
**Technical Novelty And Significance:** 3
**Empirical Novelty And Significance:** 3
**Recommendation:** 3
**Confidence:** 4

**Main Review:**

**Strengths:**

I appreciate the detailed explanation of the IRC, geometric intuition, and defining a diagnostics metric that they then applied to their examples. I knew that a challenge for this class of methods is having correlated columns and that this was a downside of having large libraries, but I didn't know what to do about it or a more thorough explanation. It's also nice when papers include motivation/intuition about their algorithmic improvements.

The experiments are able to demonstrate improvements while comparing to quite a few previous methods, including 2021 papers. I also appreciate this: "To make sure the comparison is fair, we compare our approach with the ones from literature using the data from the original authors of those approaches. Furthermore, we restrict ourselves to cases where the original authors have tuned their algorithms and present the cases as being hard ones, see table 1."

**Weaknesses:**

*Rigor of Experiments*
1. Why are there missing entries in Tables 2 & 3?

2. "During training, if Lmse on the test set does not vary anymore or if it increases, the sparsity estimator is triggered periodically." They shouldn't be using test error for this! A validation set could be used for this, though. Test sets should be held out until the very end if they are going to legitimately check for generalizability and overfitting. This raises questions about these claims:
    - "Our approach allows to recover all 4 PDEs without overfitting while the original DeepMod would for all, except for the KdV equation."
    - From the appendix: "Typically this can be diagnosed as we report (in TensorboardX) the mean square error on a test set which allows to verify if the discovered PDE generalizes well or not."

3. A major point in this paper is being able to handle higher noise-to-sample ratios (and larger libraries) than previous methods, which is a great goal. It's not clear to me how these results (Table 3) were established, though. Did you run each algorithm on the same data and see how much the noise could be increased until the wrong terms are selected? Similarly, did you gradually increase the library size? Or was this table constructed by checking those papers and seeing what level of noise and library size they reported for these particular PDEs? If the latter, that could explain why there are some missing entries. However, that weakens this claim in a few ways: (1) These six papers might have varied the details of how they set up these datasets, (2) they might have reported results with some particular level of noise and library size without trying to push their method to the limits, and (3) if these papers were trying to push their method to the limits, they might have had differing criteria about what the limit is. For example, are they satisfied with selecting the right terms, or do they also want the coefficients to be correct within some tolerance?


4. Related to the previous point, I'm concerned about this paper not reporting how accurate the coefficients are, either by reporting the learned coefficients or reporting their error. This is briefly touched on at the very end of the appendix without giving a lot of specifics. In contrast, the related papers that they cite typically report this information. Some also show how accurate the solution is if you solve the PDE with the estimated coefficients. Perhaps this algorithm chooses the correct coefficients more often than other papers but the coefficients are much less accurate, which would be a valuable trade-off to know about.


*Correctness of other claims/statements:*

5. "It is anecdotaly known in the model discovery community that the Lasso does not perform very well compared to other relaxations of the original problem, see Rudy et al. (2017), Li et al. (2019), Rudy et al. (2019) and Maddu et al. (2019), however it has never been studied why. We trace back this lack of performance to the potential variable selection inconsistency of the Lasso." I don't think it's fair to say that this has never been studied. For example, Maddu et al. (2019) says "For example, the studies in [43, 44] provide sufficient and necessary conditions, called the irrepresentable conditions, for consistent variable selection using LASSO, essentially excluding too strong correlations of the predictors in the design matrix." They then use this as motivation to use Randomized Lasso.

6. The implication of meeting the IRC (or having this paper's diagnostic metric be < 1) seems to be inconsistently described, which confused me. I think that if IRC is violated, the Lasso could still be correct, right? Then these two sentences are a little too strong.
    - "For both libraries [the diagnostics metric] > 1 revealing the Lasso would not be able to recover the true support."
    - [The diagnostic metric after adaptive Lasso is greater than 1] "explaining why the adaptive Lasso alone will not work in those cases."

7. "In addition, it allows to preserve a convex optimization problem and experimental results show it can select the true model in challenging cases at much lower computational cost than state-of-the-art approaches." This is overstated - they only show an improvement in computational cost for one example & one method.

8. This part of the proposition seems imprecise "The design matrix Θ ̃ will therefore have more chances to verify the IRC than Θ."


*Novelty:*

9. While this paper definitely contains novelty and should (with some revision) be published somewhere, I'm not sure it's enough novelty for this venue. In particular, although I definitely appreciate the deeper discussion of IRC, the Maddu et al. (2019) (PDE-STRIDE) paper also discussed it and used it for motivation to use Randomized Lasso and stability selection. This paper then adds variable selection error control and changes from Randomized Lasso to Randomized Adaptive Lasso. They then show improved results compared to PDE-STRIDE. Similarly, they show that incorporating rAdaLasso with stability selection and error control into DeepMod gives improved results over DeepMod. So, this seems worth publishing but doesn't seem like a big step in novelty to me.



**Minor feedback**, no need to address in discussion phase:
- I found it a little confusing to compare Tables 2 & 3. I'm guessing the noise level changed. Did anything else change about these test cases?
- I thought that "up to O(2) higher noise-to-sample ratios" was confusing notation. I think you want to say "two orders of magnitude," but I'm not familiar with using O(2) to say that.

- "Conversely, the Lasso will not consistently select the true variables for any design matrix, even if they are present in the library." This sentence is unclear to me. By "conversely", do they mean that without IRC, the Lasso will not consistently...?
- This sentence is hard to read: "Our approach allows to recover all 4 PDEs without overfitting while the original DeepMod would for all, except for the KdV equation."
- Figure 6 in the Appendix doesn't have much of a description, so I'm not sure what's happening.

~~~~

I should note that I did not check the proof, since it's in the Appendix.

**Summary Of The Paper:**

This paper describes an improvement to previous methods for discovering PDEs from data. They show that even for clean data, the design matrices for discovering PDEs using sparse regression can violate irrepresentability condition (IRC), which motivates their approach. They introduce randomised adaptive Lasso (rAdaLasso) and incorporate stability selection and error control. They demonstrate this approach for discovering PDEs from data that was numerically differentiated, comparing to two previous methods on two tricky PDE test cases. They also incorporate rAdaLasso with stability selection and error control into DeepMod (an existing model discovery framework that uses deep learning and avoids numerical differentiation) and show with four PDEs that this is an improvement over the original DeepMod method. Then they compare their version of DeepMod to six previous papers on two PDEs, showing the ability to tolerate more noise and larger libraries.

**Summary Of The Review:**

My primary concern is the rigor of the experiments, and my secondary concern is that some claims are overstated.

---

> ### Author Response · Authors · 2021-11-14
> **point by point**
>
> First of all we would like to thank the reviewer for her/his very insightful comments that will help without no doubt to improve our work.
> 1. see point 3
>
> 2. There is a vocabulary imprecision in our paper. When we say we use n samples for training in DeepMod, we use 80% as training data and 20% as test data - we should have called the latter validation data.
>
> 3. The table is constructed by checking the 6 papers.  Unfortunately, the authors of those papers do not all share their code and data.
>
> 4. We agree with the reviewer that we should report coefficient errors as our colleagues.
>
> 5. We agree with the reviewer, previous authors have mentionned the poor consistency of the Lasso in Li et al. (2019) and even more specifically in Maddu et al. (2019). Thankfully the authors of the latter shared both the data and code for at least one of the examples of their paper (Burgers equation), so we could test it. Our initial plan was to simply use the IHT algorithm within DeepMod - because its variable selection consistency seems empirically excellent. However, its computational cost is very high and becomes prohibitive for DeepMod: it needs to be triggered many times during the learning. This motivated digging deeper why a bare Lasso would not work and finding an alternative to the random Lasso (which does fail by the way on a hard case, see table 2). The novelty is perhaps not on the technical side but we do show the consequences of our modifications as we can solve harder problems than before.
>
> 6. Indeed, our statements are too conservative, we will revise this on the whole paper.
>
> 7. While it is true we show an improvement in only one example for the IHT algorithm, there is no reason for the computational cost to decrease by 2 orders of magnitude for any other example. All methods except the IHT in Table 2 fail in at least one hard case, so there is no need to benchmarck their computational cost in our opinion. Furthermore solving a convex optimization problem was in our minds the best warranty of a very low computational cost, explaining why we tried to push as far as possible a Lasso based solution. But we agree, it would be nice to discuss other methods.
>
> 8. We agree with the reviewer, the sentence needs revision.
>
> Overall, it seems to be an agreement among reviewers that our paper lacks novelty so we will work on this before re-submitting.

---

> > ### Comment · Reviewer_pXwU · 2021-11-21
> > **response to authors**
> >
> > Hello,
> >
> > I'm glad to hear that you are open to revising and re-submitting. I would like to echo the last reviewer in saying that I'm sure this will be a great paper in the future.
> >
> > To back up the claim about handling higher noise-to-sample ratios (and larger libraries) than other methods, it would be great if you can run at least some other methods to see how far the noise & library size can be increased. For the methods you aren't able to run yourself, you could report the noise-to-sample ratio and library size they used, but make a distinction between reporting what was done in those papers vs. pushing the methods to their limits.
> >
> > For the computational cost aspect, this statement can be revised to distinguish between the scenario that you checked versus the expectation that cost savings would also occur in comparison to other methods.
> >
> > Thanks!

---

### Decision · Program_Chairs · 2022-01-20

**Decision:**

Reject

**Comment:**

The reviewers and AC all agree that the paper considers an important problem but that several concerns remain which makes the present submission of limited novelty.
We strongly encourage the authors to revise their manuscript to incorporate the reviewers comments as this will significantly strengthen the significance of their work.
In particular it will be important to strengthen the theoretical analysis and expand the empirical evaluation, including incorporating an ablation study and considering settings of various difficulty, noise level, etc.